# Evaluation of Tropospheric Moisture Characteristics Among COSMIC-2, ERA5 and MERRA-2 in the Tropics and Subtropics

Benjamin R. Johnston *[ID], William J. Randel [ID] and Jeremiah P. Sjoberg [ID]

COSMIC Program Office, University Corporation for Atmospheric Research, Boulder, CO 80301, USA; randel@ucar.edu (W.J.R.); sjoberg@ucar.edu (J.P.S.)
* Correspondence: bjohnston@ucar.edu

**Abstract:** Global navigation satellite system (GNSS) radio occultation (RO) receivers onboard the recently-launched COSMIC-2 (C2) satellite constellation provide an unprecedented number of high vertical resolution moisture profiles throughout the tropical and subtropical atmosphere. In this study, the distribution and variability of water vapor was investigated using specific humidity retrievals from C2 observations and compared to collocated ERA5 and MERRA-2 reanalysis profiles within 40°N to 40°S from September to December 2019, which is prior to the assimilation of C2 in the reanalyses. Negative C2 moisture biases are evident within the boundary layer, so we focused on levels above the boundary layer in this study. Overall, C2 specific humidity shows excellent agreement with that of ERA5 and has larger differences with that of MERRA-2. In the tropical mid-troposphere, C2 shows positive biases compared to ERA5 (6–12%) and larger negative biases with MERRA-2 (15–30%). Strong correlations are observed between C2 and reanalysis specific humidity in the subtropics (>0.8) whereas correlations are slightly weaker in the deep tropics, especially for MERRA-2. Profile pairs with large moisture differences often occur in areas with sharp moisture gradients, highlighting the importance of measurement resolution. Locations which demonstrated weaker humidity correlations in active convection regions show that ERA5 has a negative specific humidity bias at 3 km in higher moisture environments, whereas MERRA-2 displays a large positive bias at 7 km. However, additional explanations for profile pairs with large moisture differences remain unclear and require further study.

**Keywords:** tropospheric water vapor; radio occultation; COSMIC-2; reanalysis; tropics; subtropics

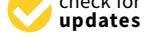

## 1. Introduction

Water vapor is the Earth's most important greenhouse gas as it accounts for nearly two-thirds of the natural greenhouse effect [1]. It plays a major role in the global energy cycle as a dominant feedback variable in association with radiative effects and moist dynamics [2], and fast-acting water vapor feedbacks constitute a strong amplification mechanism for anthropogenic climate change, which makes water vapor a key parameter for climate change analysis [3]. Water vapor also carries a large amount of latent heat which is released into the atmosphere during condensation and stored again through evaporation [4]. This results in the vertical distribution of tropospheric water vapor controlling many aspects of the climate, especially through its influence on shallow and deep convection throughout the tropics [5]. In contrast to other greenhouse gases such as carbon dioxide or methane, water vapor has a much higher temporal and spatial variability [6]. Thus, accurate and consistent tropospheric water vapor measurements are essential for studying water vapor feedbacks on the global energy budget, which is still one of the largest uncertainties in understanding global warming [7].

To address this need, atmospheric scientists have developed many tools to measure the vertical and horizontal distribution of water vapor, although each method has limitations. Historically, water vapor has been measured using radiosondes, which allow for

the direct measurement of moisture with a high vertical resolution at each launch station. However, there are only about 1000 radiosonde launch stations around the world which are sparsely distributed (e.g., mostly over land), typically launched only twice per day, and systematic bias exists depending on the type of instrumentation used [8,9]. Various water vapor remote sensing instruments have also been developed over the past 40 years, such as infrared and microwave radiometers, which allow for more complete spatial and temporal coverage than radiosondes. The drawbacks of these methods are that infrared radiometers are limited to clear-sky conditions [10] and the accuracy of microwave radiometer water vapor retrievals can be affected over different surface types due to assumptions made about surface emissivity [11]. More recently, global navigation satellite system (GNSS) radio occultation (RO) has been demonstrated to accurately measure water vapor with a high vertical resolution in all weather conditions, regardless of surface type [12–16]. However, issues can arise within the moist lower troposphere. Negative RO refractivity bias commonly occurs within the boundary layer due to a combination of various receiver tracking errors and the presence of super-refraction [17–20]. For example, negative biases in bending angles and refractivity may be caused by errors in profiles with a low signal-to-noise ratio (SNR) within the complex moist lower troposphere [20]. Additionally, super-refraction of the RO signal occurs when the vertical refractivity gradient exceeds critical refraction ($dN/dz < -157$ N-units km$^{-1}$), which often results from the combination of a sharp vertical gradient in moisture and a temperature inversion frequently observed at the top of the boundary layer [21]. When lower tropospheric refractivity is negatively biased, this results in the underrepresentation of lower tropospheric moisture in the derived humidity profiles [17,22,23]. Thus, even when combining each of these observational methods, water vapor is generally observed insufficiently compared to other meteorological parameters [6].

Atmospheric reanalysis is a useful tool that addresses some of the observational limitations. Reanalyses aim to construct a continuous and complete picture of the weather and climate by constraining evolving model forecasts after assimilating a large mixture of observations with different spatial/temporal resolutions, accuracies, and degrees of correspondence to model state variables and fluxes [24]. Reanalysis datasets have many advantages, such as global coverage, multivariable outputs, and spatial integrity [4]. These characteristics can provide great value which results in reanalyses being widely used to understand atmospheric processes. However, the degree of sophistication to which water vapor and its variability are represented depends on the reanalysis system, which observations it assimilates, which microphysical and chemical parameterizations it includes, and how those parameterizations affect water vapor distributions [25]. Additionally, reanalysis humidity records have inhomogeneity issues due to the assimilation of inhomogeneous radiosonde humidity data [26] and reanalysis water vapor information may be less reliable in areas where limited observations are available for the data assimilation, which often means humidity variables are less certain than other reanalysis outputs such as temperature or air pressure for which many more observations are available [27]. Therefore, it is important to compare water vapor observational data with independent reanalysis data in order to validate both types of datasets.

COSMIC-2 (C2), a joint Taiwan/United States six-satellite mission, is a GNSS RO mission launched in June 2019 as a follow-on to the highly successful COSMIC research mission launched in 2006 [28]. Importantly, one of the main upgrades that C2 provides over its predecessor is a higher SNR than any other RO mission to date due to the advanced receiver and high-gain antenna onboard. This higher SNR reduces the contribution of thermal noise to bending angle errors and allows for the penetration of soundings lower into the troposphere [29]. Additionally, sampling totals and density have been improved relative to the COSMIC mission. Once fully operational, the C2 satellites are expected to produce over 5000 high vertical resolution profiles of bending angles and refractivity per day in the tropics and subtropics [29]. These advancements make C2 ideal for tropical moisture research.

In this study, we validate this new C2 data by comparing C2 specific humidity (SH) profiles with profiles from two widely used atmospheric reanalysis datasets, ERA5 and MERRA-2. Our goals in this research are to assess the data quality of these datasets and determine their strengths and limitations, to quantify tropical and subtropical moisture and its variability, and to analyze when and why moisture differences can occur between the datasets. The structure of the paper is as follows: Section 2 highlights the three datasets and details the methodology used for this study; Section 3 presents the key results of the study, such as identifying the correlations between C2 and each reanalysis, quantifying the magnitude of any differences, and providing possible physical explanations as to why these differences can occur; Section 4 offers a brief discussion of the results; finally, the conclusions are provided in Section 5.

## 2. Data and Methods

### 2.1. COSMIC-2 GNSS RO Data

Level-2 C2 GNSS RO moisture soundings are obtained from the COSMIC Data Analysis and Archive Center (CDAAC) at the University Corporation for Atmospheric Research (UCAR). We use the new "wetPf2" retrieval, which provides refractivity, temperature, and moisture from near the surface to approximately 60 km. The "wetPf2" product was designed to improve upon the older "wetPrf" retrieval by using a variational regularization on the Abel inversion during the retrieval process, which enhances the RO measurement with the aid of prior information [30]. The focus of the improvement was on the lower troposphere where the Abel inversion is hampered by a larger measurement uncertainty, especially by a considerable negative measurement bias [30]. The vertical resolution of RO soundings varies from ~200 m in the lower troposphere to ~500 m in the upper troposphere [31] while the horizontal resolution is ~200 km [28]. The retrieved profiles are reported as a function of geometric height above mean sea level and the profiles are quality controlled by excluding the ones with "bad" flags (such as if the observation bending angles exceed the climatology by a specific threshold).

Figure 1 shows the C2 sampling and profile penetration to at least 1 km above the surface (including elevation), using $5° \times 5°$ grids within the tropics and subtropics from September–December 2019. C2 sampling is ideal for tropical studies, as over 400 profiles per grid box were observed around the equator during the four-month period. Sampling slowly decreases into the subtropics, with minimum sampling totals between 150–200 profiles observed around 40° latitude. Additionally, C2 profile penetration below at least 1 km is generally very good, with most regions throughout the tropics and subtropics observing a profile penetration of >70%, with some drier regions seeing a penetration of >90%. Regions that observe slight decreases in penetration are typically over the mountainous regions (e.g., the Andes and Tibet Plateau) with more severe penetration issues (<50%) occurring above the Southern Hemisphere stratocumulus cloud regions.

### 2.2. ERA5 Reanalysis Data

ERA5 is the fifth generation of atmospheric reanalysis to be produced by the European Centre for Medium-Range Weather Forecasts (ECMWF). ERA5 replaces the outgoing ERA-Interim reanalysis, which was stopped at the end of August 2019. ERA5 now provides an enhanced number of output parameters, hourly temporal resolution, a horizontal resolution of 31 km, a vertical resolution of 137 levels up to 1 Pa, and the benefit of over 10 years of model and data assimilation (DA) developments relative to ERA-Interim [32]. Note that COSMIC-2 RO soundings had not yet been operationally assimilated into ERA5 during the study period (September–December 2019).

### 2.3. MERRA-2 Reanalysis Data

MERRA-2 is the follow-on to the original MERRA reanalysis developed by the National Aeronautics and Space Administration (NASA) Global Modeling and Assimilation Office (GMAO). It is intended as an intermediate reanalysis with some important updates,

but many basic aspects of the MERRA-2 system, such as the variational analysis algorithm and observation handling, are largely unchanged since MERRA [33]. MERRA-2 provides three-hourly temporal resolution, a horizontal resolution of approximately 50 km ($0.5° \times 0.625°$), and a vertical resolution of 72 levels from the surface to 10 Pa [33]. Again, note that COSMIC-2 RO soundings had not been operationally assimilated into MERRA-2 during the study period.

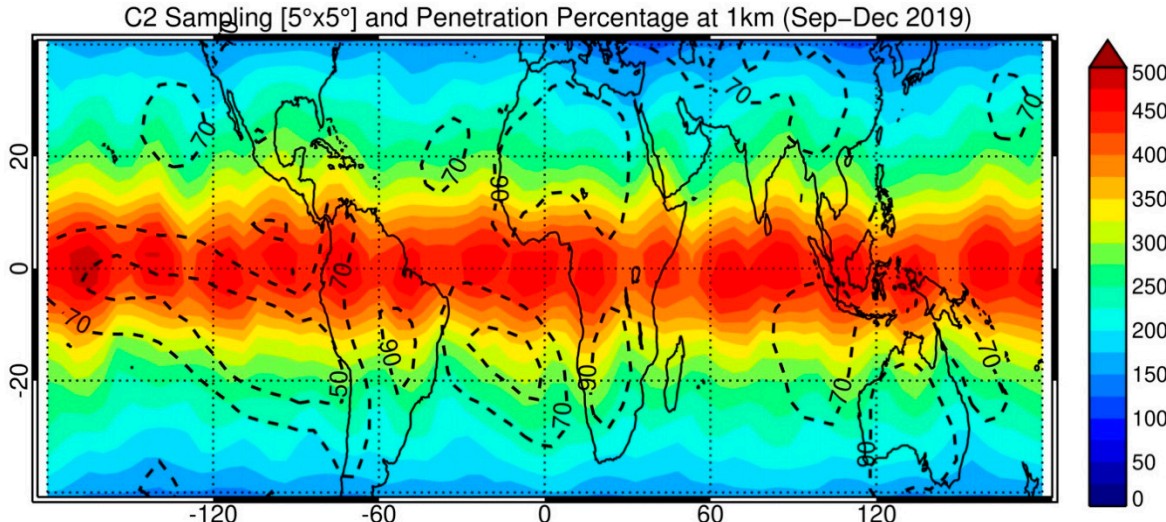

**Figure 1.** C2 sampling using $5° \times 5°$ grids (color contours) along with penetration percentage to at least 1 km above the surface (dashed contours) from September–December 2019.

### 2.4. Methodology

To observe tropical and subtropical water vapor distribution and variability and assess the quality of the new C2 dataset, C2 GNSS RO specific humidity observations, along with ERA5 and MERRA-2 reanalysis profiles, were obtained for September–December 2019 within 40°N–40°S. The gridded reanalysis data were collocated to times and locations of RO profiles, including the tangent drift of the profiles. This was done by finding the minimization of two functions: the first being the linear fit to the RO profile latitudes and longitudes dependent on the geopotential height, and the second being the bilinear fit to the model geopotential height on a given model level at a given time step dependent on latitude and longitude [34]. The minimization of these two functions returned the latitude, longitude, and geopotential heights at which the RO profile crossed the given model level at each model time step. Bilinear interpolation in space and linear interpolation in time to these crossing points gave profiles of model data on model levels that followed the path of each RO profile. The moisture profiles from all three datasets were interpolated to the same uniform 50 m vertical grid for ease of comparison.

Profile pairs (C2/ERA5 and C2/MERRA-2) were then compared and differences were recorded. Individual profile differences were displayed in scatterplots and two-dimensional histograms and were also displayed by gridding both zonally (2.5°) and by latitude/longitude ($5° \times 5°$). We focused our study on two tropospheric altitudes: 3 km was chosen as it is in the moist lower troposphere but is generally just above the boundary layer and thus avoids the negative bias commonly observed within the boundary layer in RO profiles, and 7 km was also chosen as it is within the mid-troposphere. Specific humidity differences were compared using absolute value (in g/kg) and by percentages, such that:

$$\text{SH Difference} = ((C2_{SH}\text{-REANALYSIS}_{SH})/\text{REANALYSIS}_{SH}) * 100 \qquad (1)$$

Moisture differences were also studied in smaller regions where large moisture differences and weaker correlations are observed in order to better understand why large SH differences can occur between profile pairs, including in regions that see frequent deep

convection (e.g., central Africa) or strong moisture gradients (e.g., over the Northern Pacific Warm Pool in winter).

## 3. Results

### 3.1. Tropics/Subtropics Analysis

The results of this study focus on evaluating the specific humidity differences between the C2, ERA5, and MERRA-2 datasets throughout the tropics and subtropics. First, Figure 2 shows the overall C2 specific humidity characteristics on a zonal and latitude/longitude basis to highlight the usefulness of C2 RO profiles in displaying large-scale moisture features and providing context for analyzing the moisture differences observed between datasets in the next sections. Zonal mean moisture contours (Figure 2a) show that lower tropospheric moisture amounts are highest near the surface within the tropics (~15 g/kg) and decrease quickly away from the deep tropics (6–8 g/kg near 40°) and with height (<1 g/kg into the upper troposphere). More structure is evident when analyzing moisture characteristics on a latitude/longitude basis at different levels, with various moist and dry spots apparent. At 3 km (Figure 2b), moisture amounts are highest throughout the deep tropics, with mean values near 8 g/kg found over the far eastern Pacific and Atlantic Oceans, northern South America, the northern Indian Ocean, and the Pacific Warm Pool. Subtropical SH variations are also observed, with maximums (>4 g/kg) typically occurring over the warm ocean currents (e.g., the Gulf Stream and the Kuroshio Current) and minimums (<2 g/kg) observed above the marine stratocumulus cloud regions, both of which are associated with oceanic subtropical highs. Moisture distributions are similar at 7 km (Figure 2c), although some regional differences are observed. For example, the driest areas at 3 km above the Southern Hemisphere stratocumulus regions occur within the subtropics (between 20°–30°) near the coastlines, whereas at 7 km, the driest areas are shifted away from the coastlines and closer to the equator.

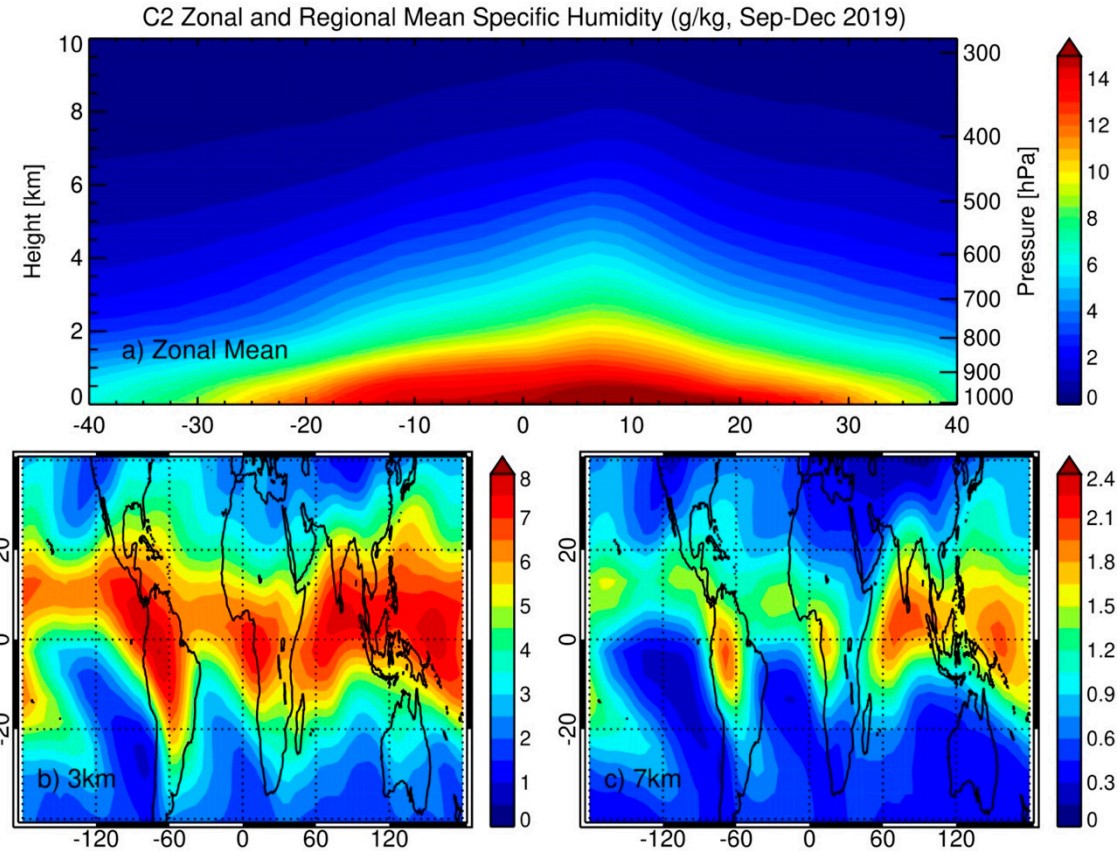

**Figure 2.** (**a**) C2 zonal mean specific humidity (g/kg), along with the C2 specific humidity at (**b**) 3 km and (**c**) 7 km within 40°N–40°S from September–December 2019.

The zonal mean humidity percentage differences between C2 and ERA5 along with C2 and MERRA-2 are analyzed between 40°N–40°S in Figure 3. Profile pairs are placed within 2.5° latitude bands and humidity percentage differences are calculated and averaged to obtain a better idea of which latitudes and tropospheric heights show the largest differences. The C2 and ERA5 comparison (Figure 3a) shows that C2 has a negative specific humidity bias within the boundary layer (–6% to –12%). These differences occur due to well-known issues with RO within the boundary layer, as large refractivity changes associated with a strong inversion layer at the top of the planetary boundary layer (PBL) can lead to super-refraction and various receiver tracking errors which cause negative biases in RO refractivity measurements [18,20]. Just above the boundary layer, C2 and ERA5 SHs show excellent agreement. As heights increase, the C2 bias becomes positive and the maximum bias (9% to 12%) is observed in the Northern Hemisphere tropics between 8 km and 9 km. C2 and MERRA-2 moisture differences (Figure 3b) within the lower troposphere are similar to those of C2 and ERA5, with negative bias observed in the boundary layer (–9% to –15%) and good agreement just above the boundary layer. However, as heights increase, SH differences increase rapidly and a large negative bias is observed, similar to Rieckh et al. [34]. Bias is especially large in the tropics (generally between –24% to –30%) and decreases into the subtropics (between –12 to –21%). The most likely explanation for these large differences in the tropical upper troposphere is due to the parameterization of large-scale condensation and evaporation within the atmospheric general circulation model (AGCM) used in MERRA-2. This parameterization scheme contains a series of new settings that result in a substantial increase in the evaporation of cloud ice crystals [35], leading to higher than expected specific humidity values throughout the upper troposphere (especially in the tropics).

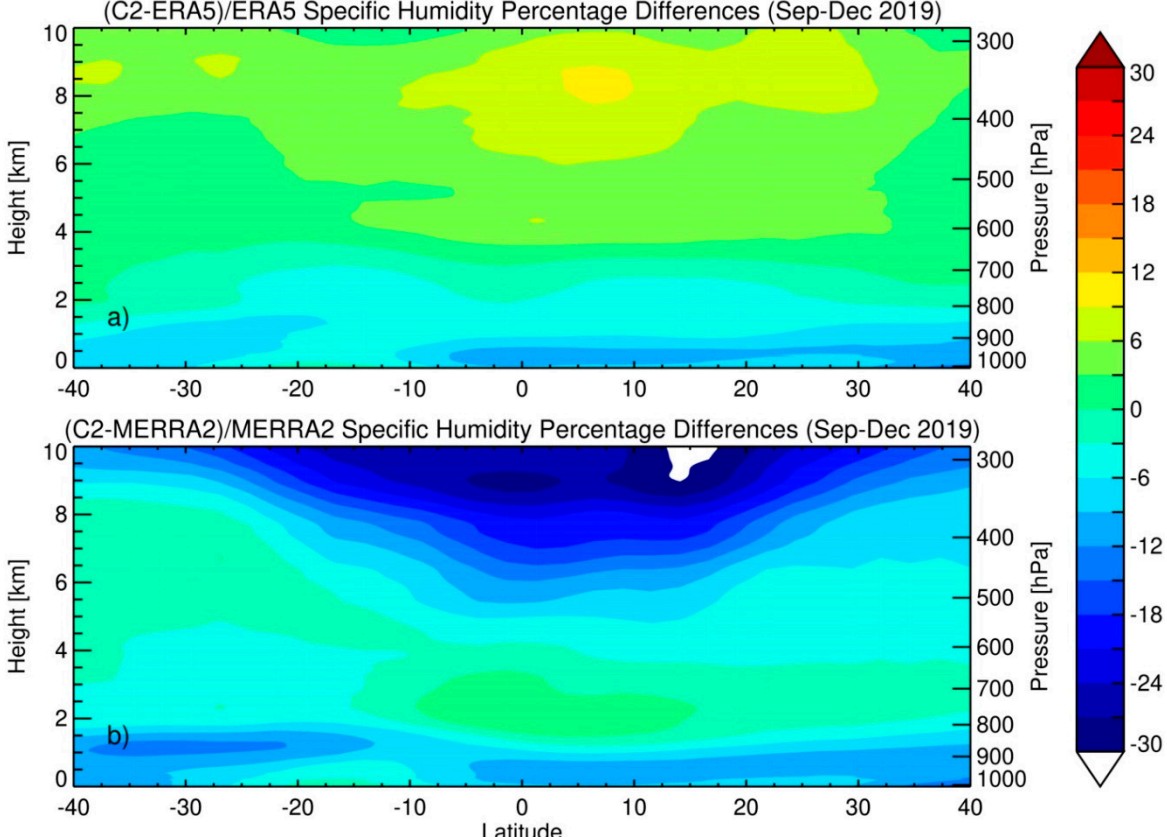

**Figure 3.** (**a**) C2 and ERA5 zonal specific humidity percentage differences, and (**b**) C2 and MERRA-2 zonal specific humidity percentage differences within 40°N–40°S from September–December 2019.

Specific humidity differences are also quantified on a latitude/longitude basis in Figure 4 to determine where the largest biases are occurring. Two tropospheric altitudes are chosen for analysis throughout the rest of this study: 3 km (~700 hPa, just above the boundary layer) and 7 km (~400 hPa, in the middle-troposphere). Minor differences are observed for C2 and ERA5 at 3 km (Figure 4a), with the largest negative bias (–3% to –9%) observed over the tropical Pacific and southern Indian Oceans and a slight positive bias (up to 6%) over portions of the Mediterranean and Asia. While zonal mean differences between C2 and MERRA-2 were minor at 3 km, large bias hotspots are observed in multiple regions (Figure 4b), especially in the Southern Hemisphere. These bias hotspots are generally within ±20%, although the positive bias within the southern Pacific Ocean is greater than 30%. Time series analysis within these bias hotspots (not shown) revealed that the bias is consistent throughout the study period. Interestingly, the hotspots have somewhat of a dipole structure over the Southern Hemisphere oceans, especially near the stratocumulus cloud regions of the Atlantic and Pacific oceans. In these regions, super-refraction of the RO signal can produce severe negative bias within refractivity profiles [21]. However, the boundary layer in these regions is typically between 1–2 km [18] so it is unlikely that super-refraction could cause these large bias hotspots at this altitude, especially as the bias is not observed when comparing C2 with ERA5. At 7 km, positive bias is observed for C2 and ERA5 (Figure 4c) within the tropics, with the largest bias (9% to 15%) occurring in regions with frequent deep convection. For C2 and MERRA-2 (Figure 4d), a large negative bias is observed within the tropics (–12% to –30%), with the largest differences (20–30%) again observed over regions with frequent deep convection. Differences become smaller in the subtropics as the Southern Hemisphere bias is between ±6%, although larger differences (–12% to –18%) can occur over the warm Gulf Stream and Kuroshio Current, possibly due to deeper clouds (and, in turn, higher SHs from cloud ice crystal evaporation in the MERRA-2 AGCM) commonly occurring over these regions.

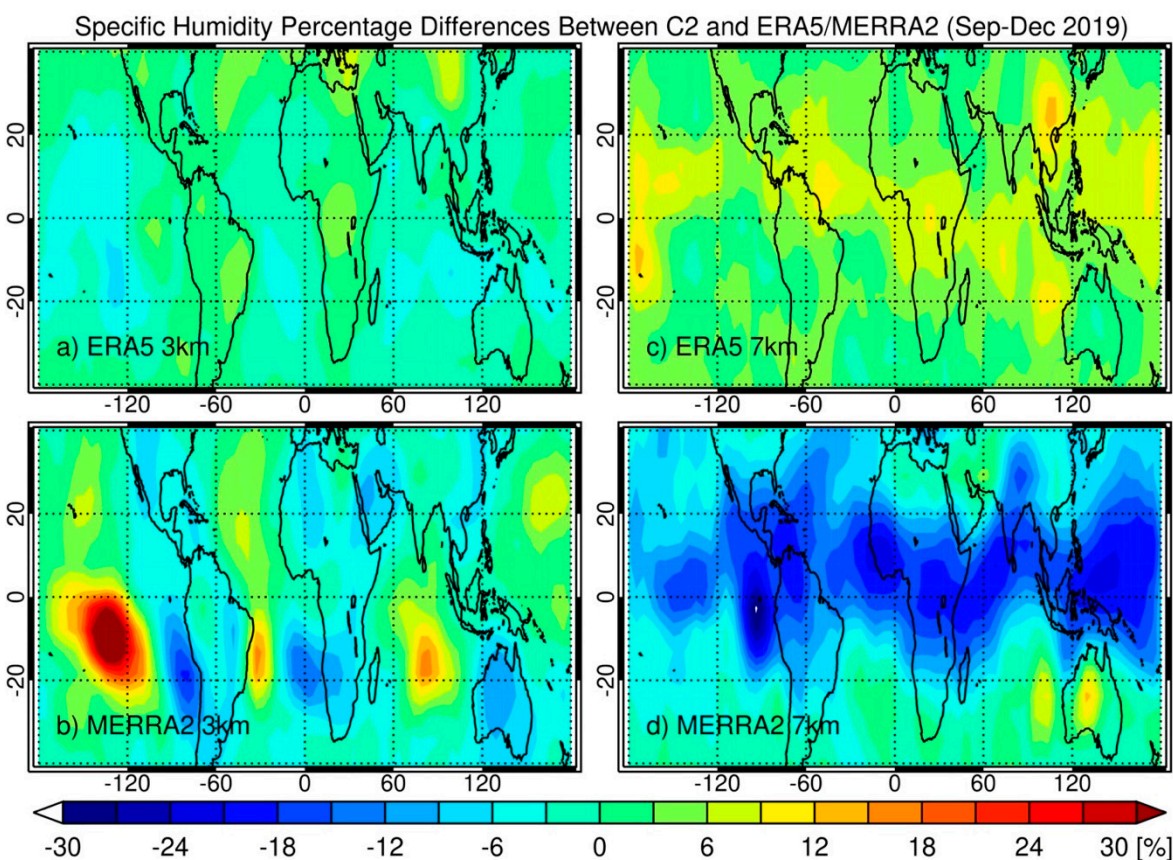

**Figure 4.** (**a**,**c**) C2 and ERA5 specific humidity percentage differences at 3 km and 7 km, and (**b**,**d**) C2 and MERRA-2 specific humidity percentage differences at 3 km and 7 km within 40°N–40°S from September–December 2019.

Figure 5 looks closer at the strength of the relationship between pairs of C2 and reanalysis SH profiles by showing contours of the gridded Pearson correlation coefficients derived from these profile pairs. The correlation coefficients are calculated using the SH values at the selected altitudes from all collocated C2 and reanalysis profile pairs within $5° \times 5°$ grids. Correlations between C2 and each reanalysis are generally strong, with the best agreement in the subtropics and slightly weaker correlations in the tropics. Correlation coefficients between C2 and ERA5 pairs are often greater than 0.9 at 3 km (Figure 5a). However, slightly poorer correlations are observed throughout the deep tropics (10°N–10°S) along with some lower correlation hotspots in regions that see frequent deep convection, such as central Africa and northern South America (which will be shown in greater detail later in the study). C2 and MERRA-2 pairs at 3 km (Figure 5b) show slightly poorer correlations throughout the entire study region, and smaller coefficients (0.4–0.7) are more consistently observed throughout the entirety of the deep tropics. For C2 and ERA5 at 7 km (Figure 5c), agreement is even better compared to 3 km, with gridded coefficients in the deep tropics now above 0.8. However, correlations are again weaker for C2 and MERRA-2 (Figure 5d), with some gridded coefficients in the equatorial Pacific smaller by 0.3–0.5. Interestingly, correlations over many of the subtropical dry regions, such as northern Africa and Australia, are poorer at 7 km relative to 3 km, but the reason for this is unclear.

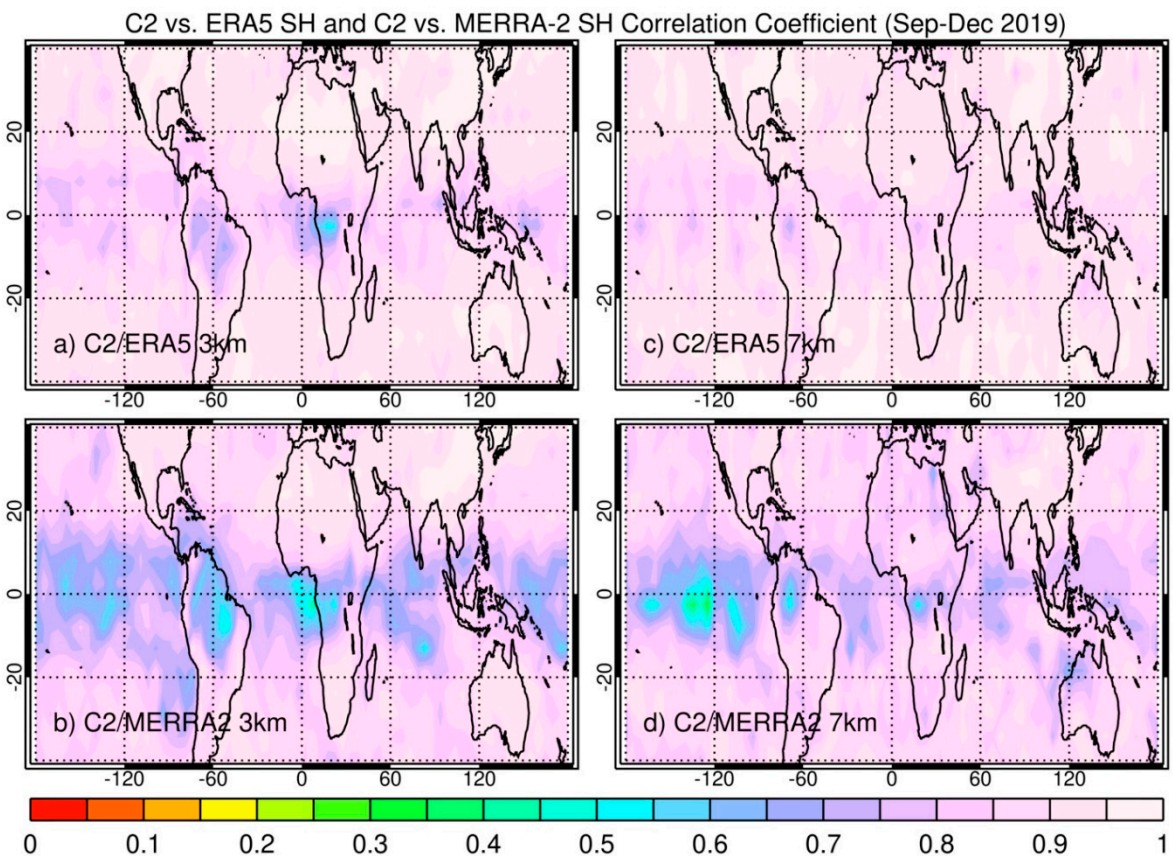

**Figure 5.** (**a**,**c**) Gridded correlation coefficients for C2 and ERA5 profile pairs at 3 km and 7 km, and (**b**,**d**) gridded correlation coefficients for C2 and MERRA-2 profile pairs at 3 km and 7 km within 40°N–40°S from September–December 2019.

Figure 6 looks more closely at the specific humidity profile pair sampling distributions within the deep tropics (10°N–10°S) due to the lower correlations observed there in the previous figure. Frequent deep convection is observed throughout the deep tropics. Since reanalysis assimilation systems often rely on moisture data sources that have larger uncertainties (or are unable to provide any moisture information) in the presence of deep convection, it is possible that this could be impacting SH variability between the profile

pairs. Overall, the agreement between C2 and both ERA5 and MERRA-2 is very good, although large differences between profile pairs are also infrequently observed. At 3 km, a strong agreement is observed between C2 and ERA5 profile pairs (Figure 6a) as most pairs are clustered along the 1:1 line and the correlation coefficient is 0.908. The correlation is slightly weaker for C2 and MERRA-2 (Figure 6b) and a larger spread is observed within the sampling distribution. For example, extreme cases show SH pair differences of up to 9 g/kg, whereas extremes are not quite as large for ERA5, and these extremes will be explored in further detail in subsequent sections. Similar patterns are observed at 7 km. Agreement between C2 and ERA5 (Figure 6c) is again better than C2 and MERRA-2 (Figure 6d), as the correlation coefficients are 0.925 and 0.859, respectively. While C2 and ERA5 agree very well when moisture amounts are low, C2 exhibits larger SH values than ERA5 when moisture amounts become higher. C2 and MERRA-2 also agree well for low SH values, but MERRA-2 generally displays much more moisture than C2 as SH values increase.

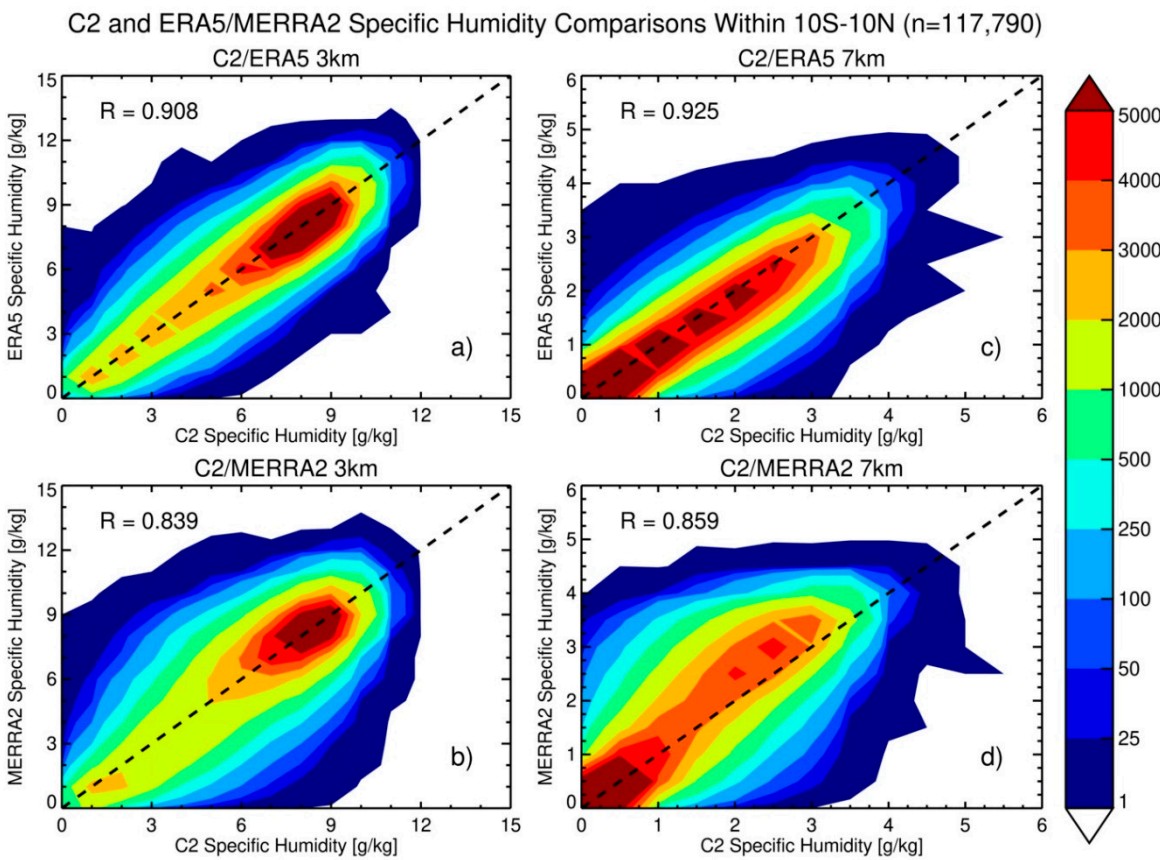

**Figure 6.** (**a**,**c**) Sampling distribution of C2 and ERA5 specific humidity (g/kg) at 3 km and 7 km, and (**b**,**d**) sampling distribution of C2 and MERRA-2 specific humidity (g/kg) at 3 km and 7 km within 10°N–10°S from September–December 2019. The number of collocated profile pairs is shown in the title and the correlation coefficient is also shown on each panel.

We also explore the relationship between the standard deviation of the SH differences between C2 and reanalysis pairs to the mean outgoing longwave radiation (OLR) from ERA5 throughout the tropics and subtropics in Figure 7. OLR has been used in many past studies as a proxy for deep convection [36–39] with lower mean OLR values (in W/m$^2$) indicating more frequently occurring deep convection (and vice versa), and we speculate that deep convection may play a role in some of the larger humidity differences observed. At 3 km (Figure 7a,b), standard deviation patterns are comparable for both C2/ERA5 and C2/MERRA-2, although variation is generally larger across all regions for C2/MERRA-2, similar to Rieckh et al. [32]. Except for the eastern Pacific Inter Tropical Convergence Zone (ITCZ) and the South Pacific Convergence Zone (SPCZ), large variability "hotspots" do not match up extremely well with lower OLR regions at 3 km. At this altitude, most

hotspots are located along the edges of low OLR regions, such as over the Gulf Stream and Kuroshio currents and over Argentina. However, deep convection occurs in these regions as well [40] and may play a role in the observed variation. At 7 km (Figure 7c,d), OLR matches up extremely well with the standard deviation of SH differences for both C2/ERA5 and C2/MERRA-2. For example, in regions with a higher mean OLR (such as over the stratocumulus cloud regions near the western coasts), standard deviations are at a minimum whereas regions with a low mean OLR (including the ITCZ, Pacific Warm Pool, and the central Indian Ocean) are at a maximum. While it is unclear why agreement is better at 7 km, the relationship between the tropical vertical water column and convection may play a role. Water vapor has much less spatial uniformity than temperature in the tropics and is more locally influenced by evaporation, precipitation, and advection [41], especially in the free troposphere well-above the boundary layer. In the tropics, the boundary layer height can often be near or even above 3 km [42,43] which could limit variability at this height. While OLR and SH difference standard deviation patterns lend support to the idea that larger moisture differences between C2 and the reanalyses can occur due to deep convection, a strong relationship was not observed between transient OLR and profile pair SH differences (not shown). Thus, the influence that deep convection may have on moisture will receive more in-depth study in future research.

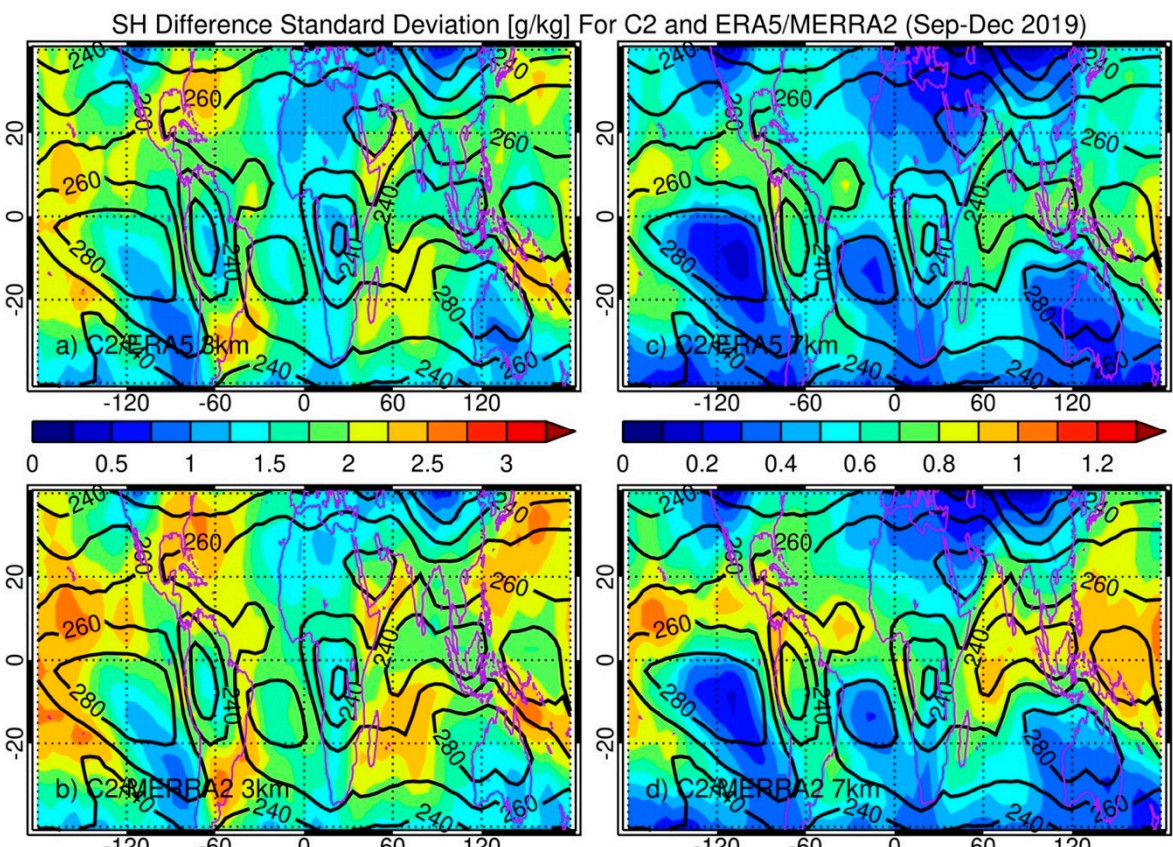

**Figure 7.** (**a**,**c**) C2 and ERA5 standard deviation of specific humidity differences (g/kg, color contours) at 3 km and 7 km, and (**b**,**d**) C2 and MERRA-2 standard deviation of specific humidity differences (g/kg, color contours) at 3 km and 7 km along with the mean outgoing longwave radiation (OLR, solid black contours) from ERA5 within 40°N–40°S from September–December 2019.

### 3.2. Regional Analysis

In this section, we shift focus from larger-scale moisture comparisons to smaller-scale regional analysis in places that see either frequent deep convection or recurring sharp moisture gradients to better understand what causes large humidity differences between profile pairs. First, the northern Pacific Warm Pool is studied as this region

has frequent occurrences of extremely dry air intrusions in the low-to-mid troposphere (3–9 km) in the winter months along with sharp moisture gradients [44]. Figure 8 shows the regional specific humidity over the Pacific Warm Pool (in g/kg) from both ERA5 (top) and MERRA-2 (bottom) at 3 km on 21 December 2019 at 12:00 UTC. A C2 profile was also observed at this time and the profile tangent points are shown by the solid white and black line. Both reanalyses show a combination of moist air (SH > 10 g/kg) and extremely dry air (SH < 1 g/kg) separated by a very sharp gradient at roughly 10°N. The lower tropospheric portion of the C2 profile is located within the sharp moisture gradient in both reanalyses, which results in large SH differences between the profile pairs. Of course, inherent differences within the model physics can generate some of the larger SH differences between profile pairs, especially in regions with sharp moisture gradients such as these. Thus, a shift in the location of either reanalysis maximum SH gradients of just 100 km can result in the C2 moisture being slightly smaller or much larger than the corresponding reanalysis SH (as shown here for ERA5). The presence of sharp gradients can also cause errors in the retrieved moisture from C2. The derivation of accurate atmospheric refractivity profiles requires precise bending angle measurements, which are only valid assuming local spherical symmetry [45]. However, as the horizontal resolution of GNSS RO is ~200 km on average, sharp lower tropospheric moisture gradients result in departures from spherical symmetry. Healy [46] produced simulations where bending angle errors near the surface exceeded 10% dependent on the horizontal gradient of the bending angle profile, which would then produce large errors in retrieved moisture. Therefore, differences in underlying model physics within the reanalyses and errors due to departures from spherical symmetry within the retrieved RO moisture profiles could result in many of the large SH differences observed.

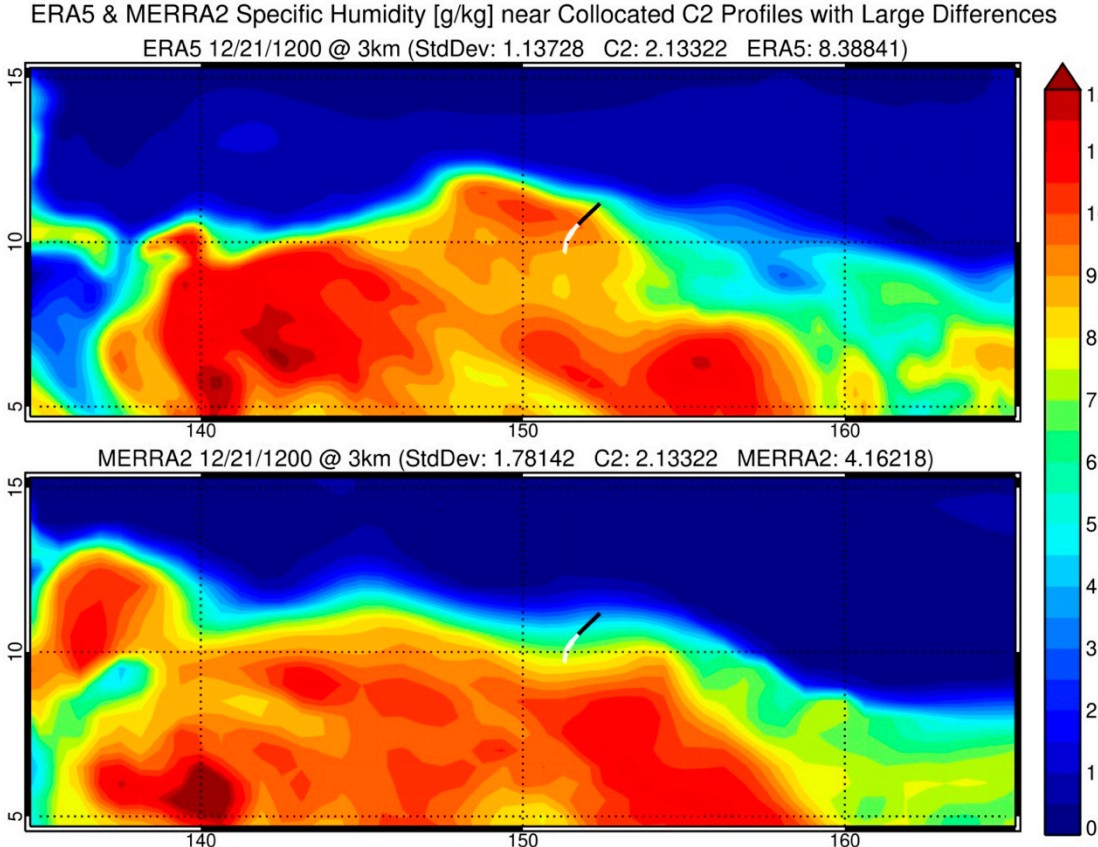

**Figure 8.** (**Top**) ERA5 regional specific humidity at 3 km and (**Bottom**) MERRA-2 regional specific humidity at 3 km on 21 December 2019 at 12:00 UTC, with a large moisture gradient present and large specific humidity differences relative to the collocated C2 profile. The collocated C2 profile tangent points are indicated by the solid white and black line, with the black line indicating tangent points below 5 km.

Next, we investigate the relationship between profile pair SH differences and the maximum local reanalysis SH gradients per 100 km near each pair in Figure 9. The profiles are obtained within 10°–15°N and 150°–155°E, which is within the same region shown in Figure 8. The local SH gradient is obtained over a roughly 300 km transect (transect lengths are slightly different due to differing reanalysis resolutions) in all directions, with only the maximum gradient saved for analysis. This transect length was chosen as it is slightly larger than the native horizontal resolution of C2. The observed SH differences are generally larger as humidity gradients become larger at both altitudes and for both ERA5 and MERRA-2. For C2 and ERA5 at 3 km (Figure 9a), a moderate relationship (R = 0.597) is observed between the two variables, as SH differences typically increase as the local humidity gradient becomes larger. The relationship between C2 and MERRA-2 at 3 km (Figure 9b) is a bit milder (R = 0.446) and the spread between points is larger, indicating larger SH differences occurring more often. Similar patterns are also observed at 7 km (Figure 9c,d). Correlation coefficients are almost the same (0.413 and 0.426, respectively) albeit with slightly less obvious trends. Based on the sample size (*n* = 251), each of these correlation coefficients are significant at the 99% confidence interval. There are some instances where maximum gradients are relatively large and SH differences are small. In these cases, SH differences can be affected by the location of the C2 ray path relative to the gradient (e.g., if both the C2 ray path and the ERA5 profile are on the moist side of the gradient). Additionally, the direction of the C2 ray path relative to the SH gradient was analyzed (not shown). Larger SH differences were more commonly observed when the ray path crossed through the moisture gradient (e.g., Figure 8), although large differences were seen even when the ray path was parallel to the isohumes.

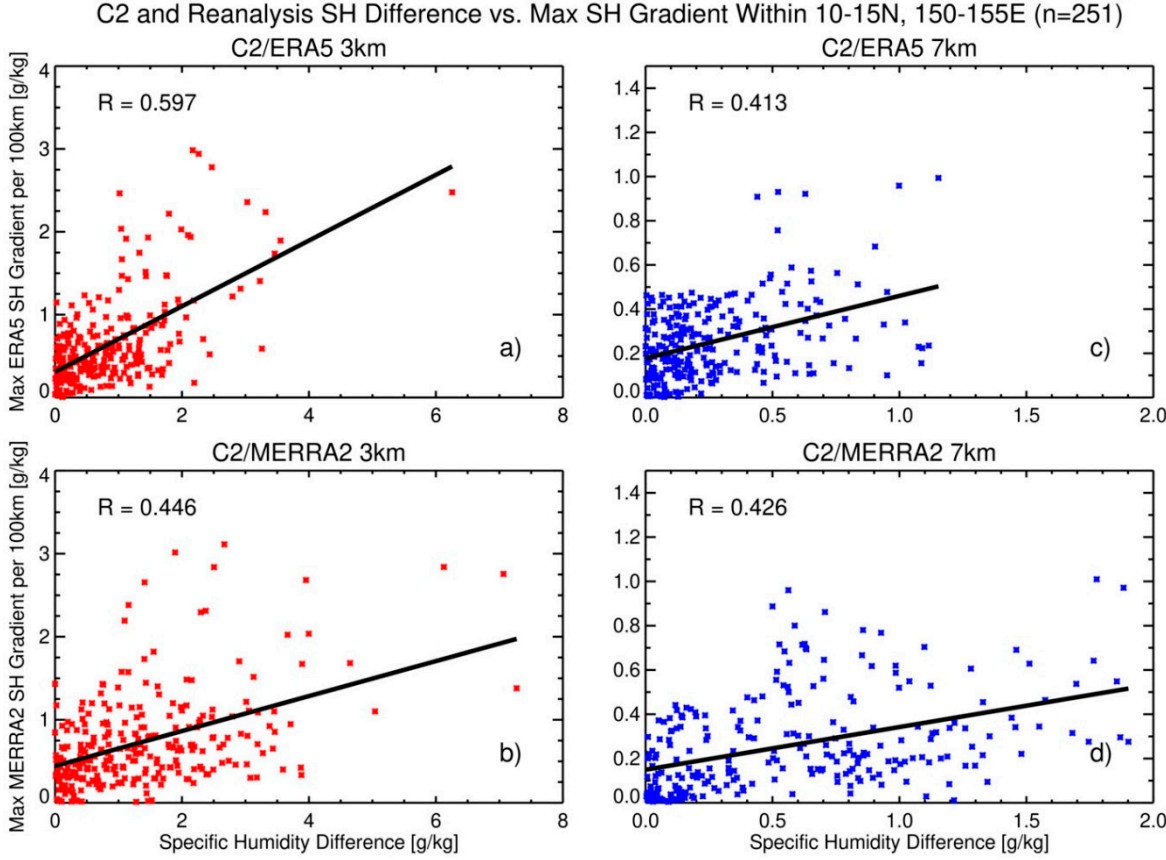

**Figure 9.** (**a**,**c**) Scatterplot of C2 and ERA5 specific humidity differences vs. the ERA5 local maximum moisture gradient per 100 km at 3 km and 7 km, and (**b**,**d**) C2 and MERRA-2 specific humidity differences vs. the MERRA-2 local maximum moisture gradient per 100 km at 3 km and 7 km within 10°N–15°N, 150°E–155°E from September–December 2019. The number of collocated profile pairs is shown in the title and the trend line and correlation coefficient are also shown on each panel.

The last analysis section focuses on two regions with poorer correlations from Figure 5. First, we look more closely at central Africa, which is a region that has frequent convection along with some of the deepest convection in the world [47]. This small region was the only location at either altitude where the C2 and ERA5 correlation coefficient was <0.65. Figure 10 shows C2 and ERA5/MERRA-2 SH profile pair sampling distributions at 3 km and 7 km within 5°S–5°N, 20°E–30°E. At 3 km, agreement is generally better between C2 and MERRA-2 (Figure 10b) than C2 and ERA5 (Figure 10a) even though coefficients are similar. This better agreement is especially obvious in higher-moisture environments. For example, when specific humidity is between 7–10 g/kg, the C2 and MERRA-2 sampling distribution is centered along the 1:1 line whereas ERA5 displays a negative bias of nearly 0.5 g/kg. However, when moisture amounts become lower (between 5–7 g/kg), agreement between C2 and each reanalysis is relatively similar. At 7 km, correlation coefficients become closer to 1 for both reanalyses. Agreement between C2 and ERA5 (Figure 10c) is excellent in low SH environments, while a negative bias for ERA5 SH appears as moisture begins to increase and is largest in the highest SH environments. In contrast, a large positive bias for the MERRA-2 SH is observed at all humidity values (Figure 10d) and confirms the results shown in Figures 4 and 6.

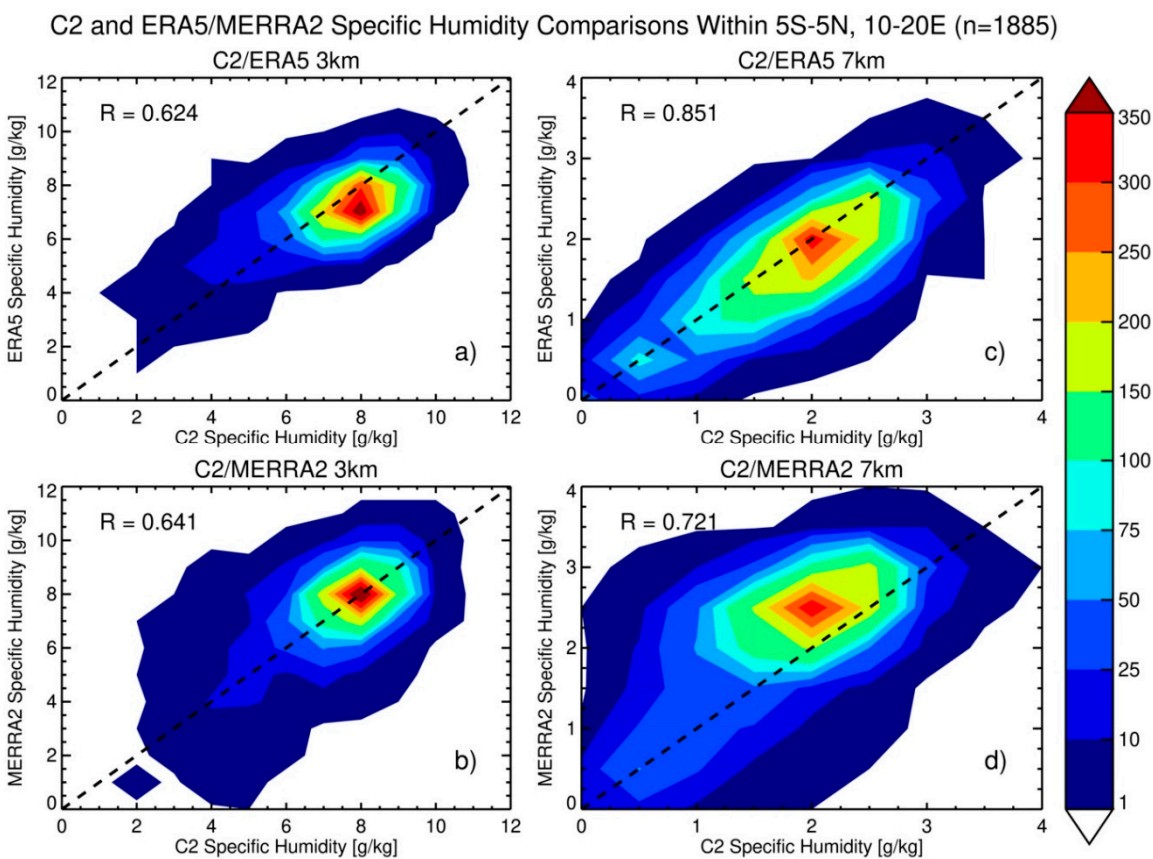

**Figure 10.** (**a**,**c**) Sampling distribution of C2 and ERA5 specific humidity at 3 km and 7 km, and (**b**,**d**) sampling distribution of C2 and MERRA-2 specific humidity at 3 km and 7 km within 5°S–5°N, 10°E–20°E from September–December 2019. The number of collocated profile pairs is shown in the title and the correlation coefficient is also shown on each panel.

Finally, we look at northern South America, which is another region with poorer correlations between C2 and reanalysis SH and also sees frequent convection. Figure 11 shows C2 and ERA5/MERRA-2 SH profile pair sampling distributions at 3 km and 7 km within 5°S–5°N, 60°W–70°W. At 3 km (Figure 11a,b), the C2 and ERA5 correlation coefficient increases considerably (R = 0.754) relative to over central Africa (R = 0.624). While C2 still displays slightly more moisture than ERA5 at high moisture values (e.g., >8 g/kg), better agreement is generally observed relative to over central Africa. On the other hand, poorer

agreement is seen for C2 and MERRA-2 over northern South America compared to central Africa, which results in slightly weaker correlations (R = 0.601 vs. R = 0.641). However, even though the spread is large, the humidity contours are still centered along the 1:1 line for high moisture values. At 7 km (Figure 11c,d), C2 and ERA5 again display excellent agreement, with even slightly better agreement at high moisture values relative to central Africa. However, C2 and MERRA-2 again have slightly weaker correlations (R = 0.679 vs. R = 0.721) and a larger positive bias is observed for MERRA-2. The number of profile pairs where MERRA-2 has much more moisture than C2 is surprisingly high. For example, there are roughly 50 pairs where the MERRA-2 SH is ~3 g/kg and C2 SH is only 1.25 g/kg, which is even more than what was observed over central Africa.

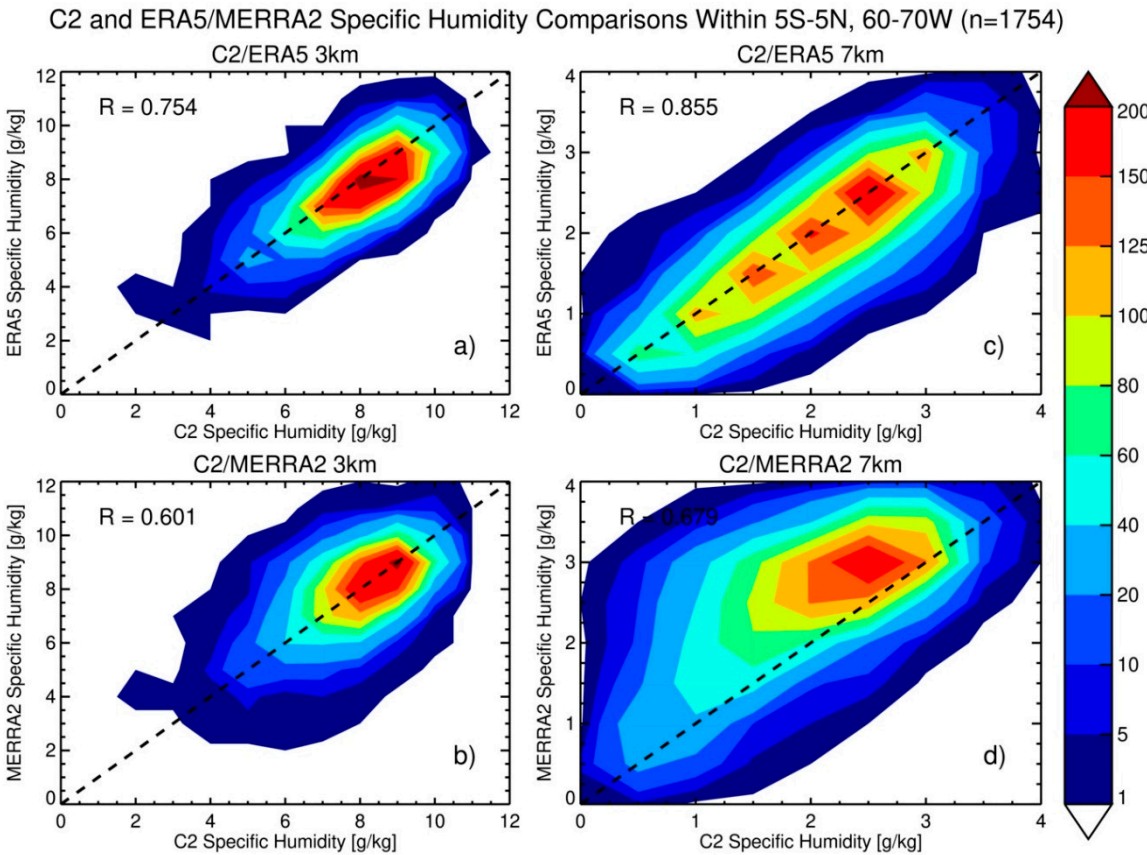

**Figure 11.** (**a**,**c**) Sampling distribution of C2 and ERA5 specific humidity at 3 km and 7 km, and (**b**,**d**) sampling distribution of C2 and MERRA-2 specific humidity at 3 km and 7 km within 5°S–5°N, 60°W–70°W from September–December 2019. The number of collocated profile pairs is shown in the title and the correlation coefficient is also shown on each panel.

## 4. Discussion

This research shows that C2 moisture information, in general, agrees very well with ERA5 and has larger differences with MERRA-2 (although it still shows relatively good agreement). C2 moisture shows a negative bias within the boundary layer, which is similar to previous research using C1 data [16,22,23]. Agreement between C2 and both reanalyses was closest just above the boundary layer (3 km) and decreased into the mid-troposphere (7 km), with C2 displaying slightly more moisture than ERA5 and much less moisture than MERRA-2. Strong correlations were displayed between C2 and both reanalyses, although correlations were typically stronger for C2 and ERA5. Correlations were generally strongest in the subtropics towards the midlatitudes and a bit weaker in the deep tropics. On first glance, it seems this could be related to overall moisture content, as higher SH is observed in the tropics and lower SH is observed in the subtropics. However, this does not hold true everywhere. For example, excellent correlations (>0.9) were observed in subtropical regions with more moisture (such as in the southeastern United States and southeast Asia) whereas

poor correlations were observed in dry tropical regions (such as over the stratocumulus regions off the west coast of South America). Thus, more work is needed to determine the cause of the larger SH variability between C2 and each reanalysis in the deep tropics.

However, even though C2 and reanalysis SH agreement is generally good, collocated profile pairs can still show large differences. These large SH differences were observed regardless of whether C2 was compared with ERA5 or MERRA-2, although perhaps unsurprisingly, more variability was observed between C2 and MERRA-2 likely due to the coarser resolution (both spatial and temporal) of MERRA-2, as well as the underlying AGCM used. Additionally, there were a non-insignificant number of profile pairs that exhibited extremely large moisture differences (e.g., greater than $\pm 4$ g/kg). For example, in Figure 2, the maximum regional mean specific humidity at 3 km is near 8 g/kg. Thus, profiles pairs with SH differences of over 4 g/kg result in biases of 50% or more. Scatterplots of C2-ERA5 humidity vs. C2-MERRA-2 humidity were also made (not shown) which showed that when larger differences between C2 and one reanalysis do occur, they are not necessarily associated with large differences between the corresponding pairs of C2 and the other reanalysis. This indicates that model resolution and physics likely play a large role when large differences do occur, and this would be exacerbated in locations experiencing sharp humidity gradients.

Additionally, the largest moisture differences and weakest correlations were typically observed in regions that experience frequent convection. For example, the largest positive bias between C2 and ERA5 at 7 km was observed along the ITCZ, over the Pacific Warm Pool, or over central Africa. We speculate that this could be due to the underrepresentation of moisture within ERA5 in the presence of convection (e.g., deep clouds), but this is purely speculation and would need further studies to confirm. Time series analysis of C2 and reanalysis SH differences were conducted over central Africa (not shown) which displayed a distinct shift in the trend of the differences in late November. For example, C2 exhibited relatively good agreement with MERRA-2 at 3 km from September through late November which shifted to C2 consistently observing less moisture from late November through December. We surmise that the observed shift in average moisture differences in late November is caused by changes in the location of the ITCZ in this region. Lashkari et al. [48] showed that the mean position of the ITCZ is typically on the northern edge of this region at the beginning of September. This location starts slowly trending farther south in October and November, and a large shift occurs in December, with the mean position of the ITCZ cutting through the region. Interestingly, MERRA-2 also displayed a large positive SH bias in active deep convection regions at all humidity values at 7 km (e.g., Figures 10d and 11d). It was relatively surprising that this substantial positive bias also occurred when moisture was rather low for this region (e.g., less than 1 g/kg), as the large bias within the tropics is due to increased evaporation of ice crystals within deeper tropical clouds in the MERRA-2 AGCM that results in higher humidity values. However, deep clouds and convection would likely occur less frequently in these drier conditions. Thus, it is possible that additional causes for this large positive moisture bias may exist within the larger MERRA-2 framework and would likely be a beneficial topic of future research.

## 5. Conclusions

In this study, tropical and subtropical water vapor distribution and variability was explored from September–December 2019 using SH profiles from the recently launched COSMIC-2 satellite constellation and compared with profiles from the ERA5 and MERRA-2 reanalyses to determine the strength of the relationship between the datasets, quantify the magnitude of any differences, and provide possible physical explanations as to why these differences occur.

Good agreement is generally seen between C2 and each reanalysis, although larger differences are observed for the MERRA-2 time-means. On a zonal scale, C2 displayed a negative moisture bias within the boundary layer likely due to a combination of various

receiver tracking errors and the presence of super-refraction. In the mid-troposphere, C2 displayed slightly more moisture than ERA5 (6–12%) and much less moisture than MERRA-2 (15–30%), which was attributed to the underlying MERRA-2 AGCM that produces too much evaporation of tropical middle and upper tropospheric cloud ice crystals [35]. In the latitude/longitude analysis, excellent agreement was seen between C2 and ERA5 at 3 km. At 7 km, a positive bias was observed in the percentage differences, especially within the tropics (9–15%). For C2 and MERRA-2 at 3 km, large difference hotspots (±20%) were observed which had a dipole structure over the Southern Hemisphere oceans near the stratocumulus cloud regions of the Atlantic and Pacific oceans. At 7 km, a large negative bias was observed within the tropics, with the largest differences (20–30%) observed in regions that experience frequent deep convection.

Strong relationships between C2 and reanalysis SHs were observed. Pearson correlation coefficients were typically over 0.9 in the subtropics. Slightly lower coefficients were observed throughout the deep tropics, with stronger relationships observed for C2 relative to ERA5 (0.7–0.8) than MERRA-2 (0.5–0.7). A closer look at the deep tropics revealed excellent agreement between C2 and ERA5/MERRA-2 at 3 km, although larger variability was observed for C2/MERRA-2. At 7 km, C2 started to exhibit larger SH values than ERA5 when moisture amounts became higher whereas, interestingly, MERRA-2 displayed positive moisture biases in both wetter and drier conditions. The relationship between OLR and the standard deviation of the C2/reanalysis SH differences was also explored. The standard deviation of the differences was generally larger for C2 and MERRA-2. Locations of high/low OLR matched up extremely well with small/large SH difference standard deviations at 7 km and only moderately well at 3 km. While this suggests that deep convection can impact moisture differences between C2 and reanalysis, a strong relationship was not observed between transient OLR and profile pair SH differences (not shown).

Larger C2 and reanalysis moisture differences were observed in locations with strong moisture gradients. A late December Pacific Warm Pool case study was shown with a very strong moisture gradient and a C2 profile located within the gradient. This resulted in large C2 and reanalysis SH differences which highlighted the importance of resolution differences between each dataset. After examining the relationship between C2 and reanalysis profile pair SH differences and the maximum local reanalysis SH gradient, larger profile pair SH differences were generally observed as local humidity gradients became larger at both altitudes. Larger SH differences were also more commonly observed when the C2 ray path crossed through (rather than paralleled) sharp moisture gradients (not shown). Finally, we examined two regions (central Africa and northern South America) that had weaker correlations and experience frequent deep convection. The sampling distributions of profile pairs showed that ERA5 has a negative SH bias at 3 km in higher moisture environments. However, at 7 km, MERRA-2 displayed large positive bias whereas C2 agreed well with ERA5.

Overall, in spite of some profile pairs showing large differences, C2 moisture profiles generally agree well with both ERA5 and MERRA-2 moisture, which confirms the usefulness of C2 as an independent observational dataset for tropical and subtropical tropospheric water vapor research. While the time period chosen in this study is relatively short, neither reanalysis dataset had yet assimilated C2 RO observations. Thus, it is probable that C2 data assimilation would improve the moisture output of both reanalyses. Future studies are likely to look more closely at C2 water vapor retrievals in the presence of deep convection and heavy precipitation to better quantify how tropospheric moisture is impacted by convection throughout the tropics.

**Author Contributions:** Conceptualization, B.R.J. and W.J.R.; methodology, B.R.J., W.J.R. and J.P.S.; software, B.R.J.; validation, B.R.J., W.J.R. and J.P.S.; formal analysis, B.R.J. and W.J.R.; investigation, B.R.J.; resources, B.R.J.; data curation, B.R.J. and J.P.S.; writing—original draft preparation, B.R.J.; writing—review and editing, B.R.J., W.J.R. and J.P.S.; visualization, B.R.J. All authors have read and agreed to the published version of the manuscript.

**Funding:** This research was funded by the National Science Foundation through grant AGS-1522830.

**Data Availability Statement:** Publicly available datasets were analyzed in this study. COSMIC-2 data is available at https://data.cosmic.ucar.edu/gnss-ro/cosmic2/nrt/ (accessed on 25 February 2021). ERA5 data is available at https://cds.climate.copernicus.eu/cdsapp#!/search?type=dataset (accessed on 25 February 2021). MERRA-2 data is available at https://disc.gsfc.nasa.gov/datasets?project=MERRA-2 (accessed on 25 February 2021).

**Acknowledgments:** The authors would like to thank John Braun, Bill Schreiner, and the COSMIC neutral atmosphere science team for many helpful discussions on the research.

**Conflicts of Interest:** The authors declare no conflict of interest and the funders had no role in the design of the study; in the collection, analyses, or interpretation of data; in the writing of the manuscript, or in the decision to publish the results.

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
