# Peer review of "Evaluation of Tropospheric Moisture Characteristics Among COSMIC-2, ERA5 and MERRA-2 in the Tropics and Subtropics"

_remotesensing, doi:10.3390/rs13050880_

Round 1

Reviewer 1 Report

This article compares the moisture profiles from COSMIC-2, MERRA2, and ERA5.  I found this article to be exceptionally clear and well written.  The results were well explained and interesting in terms of how each set of measurement pairs differed at two very different pressure levels.  The discussion and the statistics were clear and straightforward.   Conclusions and speculations were well supported and reasonable.  I recommend publication in the present form.

Author Response

Thank you very much for reviewing the paper and for the comments, they are much appreciated.

Reviewer 2 Report

This work is overall satisfying and the analyses are comprehensive. I would recommend the publication of this paper after minor revisions. My comments and questions are described below.

1. L47: ‘only about 850 stations’. According to IGRA (https://www.ncdc.noaa.gov/data-access/weather-balloon/integrated-global-radiosonde-archive), there are currently about 1000 stations are reporting data. Please check the number.

2. L68: ‘and a homogeneous record’. The humidity record in most current reanalyses actually has inhomogeneity issue due to the assimilation of inhomogeneous radiosonde humidity data. There are some publications that have reported this issue in reanalyses, e.g., Zhang et al. (2018).

Zhang W., Lou Y., Huang J., Zheng F., Cao Y., Liang H., Shi C., Liu J., 2018: Multi-scale Variations of Precipitable Water over China based on 1999-2015 Ground-based GPS Observations and Evaluations of Reanalysis products. Journal of Climate, 31, 945-962.

3. I would like to suggest the author putting the number of matching pairs in all correlation plots.

Author Response

Please see the attached response.

Reviewer 3 Report

The article is very clear, and I did not find much to criticize about this manuscript. In the following I give a list of comments and minor errors.

Line 31: Water vapour is the most important greenhouse gas, not one of many.

Line 142: Express the horizontal resolution also in km.

Line 202: Explain the abbreviation "PBL".

Lines 283 and 284: I see here a contradiction. If the bias is independent of the moisture, why does it increase with specific humidity?

Line 296: Explain the abbreviation "SPCZ".

Line 321: Delete "next".

Figure 8: The vertical scale is not defined. Does the plot cover the range +-1° or +-10°?

Lines 358 and 359: It is not clear what a "moderate relationship" is. R = 0.597 can mean anything, it depends on the size of the sample you got. It should therefore be given as well.

Line 407: Giving three significant digits for the correlation coefficient suggests that these values are known with very high accuracy. The authors should give the uncertainty of these values. Otherwise the reader cannot judge whether MERRA-2 is really worse then ERA5. 

Lines 502 - 506: I suggest to split this sentence in two, because it is a bit difficult to comprehend. 

Author Response

Please see the attached response.

Reviewer 4 Report

Please see attached minor comments.

Author Response

Please see the attached response.
